# Sudanese Arabic Dialect Encoding Using XLM-RoBERTa Language Model: Zol-RoBERTa

## Abstract

XLM-RoBERTa has proven to be very efficient at Natural Language Understanding (NLU), as it allows to achieve state-of-the-art results in most NLU tasks. In this work we aim to utilize the power of XLM-RoBERTa in Sudanese Arabic dialect. We collected over 6 million sentences in Sudanese dialect and used them to resume training of the pre-trained XLM-RoBERTa, as it was trained on 2.5T of data across 100 languages data filtered from Common Crawl. Our model - Zol-RoBERTa- is expected to achieve better performance on Sudanese Sentiment Analysis, this clarifies that Zol-RoBERTa will work better in understanding Sudanese Dialectic, which is the domain we are targeting.

## 1 Introduction

Natural Language Processing (NLP) has witnessed significant advancements in recent years, with most studies focused on English or Latin datasets. Despite the fact that the Arabic language has gained a lot of attention recently, the colloquial dialects, such as the Sudanese dialect, remain underrepresented in the field. The Sudanese dialect is spoken by more than 40 million people,and despite its growing use on social media platforms, it is one such dialect that has received little attention in NLP research. In this context, sentiment analysis (SA) is increasingly important to gauge public opinion and reactions towards major events. While SA research for English is well-established, research on the Arabic language requires substantial effort to achieve comparable levels of performance. However, the availability of freely accessible Arabic datasets for SA is limited in terms of number, size, availability, and dialect coverage. Moreover, most available resources and research publications in Arabic SA are devoted to Modern Standard Arabic (MSA) rather than dialects

To adress this gap, we propose Zol-RoBERTa, a Sudanese language model pre-trained from XLM-RoBERTa (Conneau et al., 2019), which we apply to the Sudanese Corpus to establish sentiment classification. Our model builds on recent efforts to develop SA systems for Arabic, with a focus on the Sudanese dialect. We argue that our model can match or even outperform existing post-BERT models in various NLP tasks. Our work highlights the importance of developing language models for underrepresented dialects, such as the Sudanese dialect, to advance the field of NLP and improve the accuracy of SA for non-MSA Arabic dialects

## 2 Methodology

### 2.1 Data collection

The Sudanese dataset was collected from public telegram channels and Twitter. We collected about 13 million Sudanese sentences.

**Preprocessing the data:** We cleaned the data from all non- Sudanese Arabic syntax. We removed all symbols - , # , ? , [ , ! ...etc. - and emojis. We removed sentences containing less than 6 words.

After the cleaning step was completed, we ended up with around **6.5 million** cleaned -pure- Sudanese sentences.

## 2.2 PRE-TRAINING

We are using XLM-RoBERTa model that was trained on 2.5T of data across 100 languages and it contains about 28GB of Modern Standard Arabic Data. We then pre-trained the model on our Sudanese data to ensure the model has representation of the Sudanese Arabic dialect.

## 2.3 FINE-TUNING

The language model is fine-tuned with a classification layer for the target task. Our end-task here is Fine-tuning the model on Sentiment-Analysis on Sudanese Arabic dialect. The dataset used for the Sentiment Analysis task consists of the opinions of people on Twitter about the telecommunication services provided in Sudan (Ismail et al., 2018). As well as transportation mobility services (Abuuznien et al., 2020).

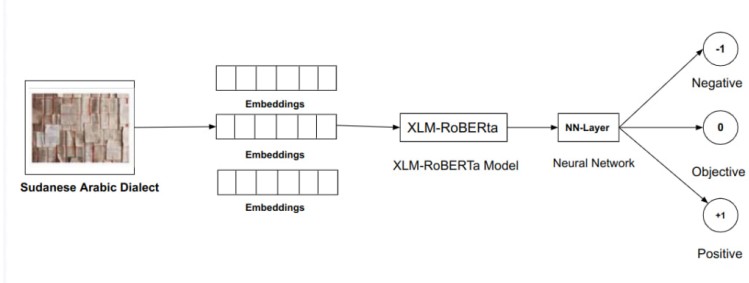

Figure 1: Steps of Fine-tuning XLM-RoBERTa for Sentence Classification.

## 3 EXPERIMENTS

We conducted experiments in google colab using Tesla T4 GPU. We used files containing 10k-13k Sudanese sentences. The pre-training was carried out with a batch size of 16 sentences per input and a learning rate of 4.88e-5. The model was trained for 25 epochs and it converged after 21275 steps, equivalent to 3.9 hours of training time. We then tested our model with huggingface Transformers library "fill-mask" pipeline to do inference with our masked language model. We provide masked text in Sudanese dialect and it returned a list of possible mask values ranked according to the score. This ensures that our model has a statistical understanding of the Sudanese dialect and can be further fine-tuned to solve different tasks, such as Sentiment Analysis.

## 4 RESULTS

Following up on our experiments. We pre-trained our model using huggingface "xlm-roberta-base" checkpoint. Due to the lack of available resources we trained the model for 1 epoch with batch size 16 and learning rate 1e-5. We then fine-tuned our model for the sentiment classification task. Our proposed model Zol-RoBERTa achieved an accuracy of 77.5% for the sentiment classification task. We will follow up on our work using Google Cloud resources and train the model for more epochs and expect to achieve higher results.

## 5 CONCLUSION AND FUTURE WORK

In this study, we present Zol-RoBERTa, a Sudanese dialect model designed for sentiment analysis. We collected and preprocessed data from Twitter and public Telegram channels to train our model using XLM-RoBERTa as a checkpoint. Our experimental results demonstrate that Zol-RoBERTa achieved high performance when dealing with Sudanese dialect, which shows promise for improving sentiment analysis in the region. As future work, we plan to compare Zol-RoBERTa's performance with SudaBERT (Elgezouli et al.) to further establish our model's state of the art results.

## URM STATEMENT

The authors acknowledge that at least one key author of this work meets the URM criteria of ICLR 2023 Tiny Papers Track."

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

## A APPENDIX

- The dataset used for the Sentiment Analysis task consists of the opinions of people on Twitter about the telecommunication services provided in Sudan (Ismail et al., 2018). It contains 4,712 tweets written in Sudanese Arabic dialect. The tweets were classified into negative (3,358), positive (716) and objective (638). As well as transportation mobility services(Abuuznien et al., 2020). The data contains 2125 tweets classified into negative, positive and objective .

- **Hyperparameters** : we use a batch size of 16 for all our models , Learning rate of 1e-05 and 3 epochs for training

