# OpenReview forum: "SUDANESE ARABIC DIALECT ENCODING USING XLM-RoBERTa LANGUAGE MODEL: Zol-ROBERTA"
_ICLR.cc/2023/TinyPapers — Submitted to Tiny Papers @ ICLR 2023_

### Official Review · Reviewer_z3DW · 2023-03-28

**Confidence:** 1

**Summary Of Contributions:**

The paper presents a solution of Sudanese dialect language analysis through encoding of the language where they have used 6 million collected sentences.

**Rating:**

Clear, Correct, and Reproducible (CCR): a submission which meets the reviewing criteria

**Strengths And Weaknesses:**

**Strength**\
the paper provide solution to Arabic language understanding through the use of transformers especially encoding through using large collected data.\
**Weakness**
*  I wonder upon the use XLM-RoBERTa doesn't the paper use the pretrained model encoders where I assume the study performed transfer learning instead of encoding.
* In order to use XLM-RoBERTa, tokenizers must be used, so I wonder if most of the tokens weren't transformed into [unk] tokens? as happens to most of pretrained models for a NLP task.

**Suggested Changes:**

Convincing explanation of difference between transfer learning and encoding where I believe the study performed transfer learning using XLM-RoBERTa

---

### Official Review · Reviewer_HNF7 · 2023-03-30

**Confidence:** 5

**Summary Of Contributions:**

The paper presents a Sudanese-Arabic dialect model XLM-RoBERTa LM (Zol-RoBERTa). The model is trained on Sudanese-Arabic dialect.

**Rating:**

Needs Clarification (NC): a submission which does not meet the reviewing criteria and needs clarification for its described problem or solution

**Strengths And Weaknesses:**

Strengths

1. The model is trained on the Sudanese dialect spoken by 40 million people.

Weaknesses

1. Novelty is weak. Previous work (SudaBERT) exists on this dialect. The authors pretrained with MLM and fine-tuned on the sentiment analysis task using standard method proposed on BERT paper, with no new methods.
2. Methodology is weak. The model is trained only for 1 setting from pre-existing "xlm-roberta-base" checkpoint. There are no other ablations experimented with that would suggest specific properties of the model.
3. Evaluation is weak. The model is only evaluated on one sentiment classification task and reports Accuracy.
4. Punctuations are sometimes missing, regarding quotation marks or end of sentences.
5. There are no citations for other previous multi-lingual fine-tuning methods which the paper is inspired from.

**Suggested Changes:**

1. The comparison between SudaBERT should be done in this paper, not left as future work.
2. Please evaluate the model on at least 2 ablations.
3. Please evaluate the model on at least 2 tasks.
4. Please revise the paper with grammatical tools.
5. Please add citations for other previous multi-lingual fine-tuning methods.
6. Please clarify the choice of the XLM-Roberta model, in terms of tokenization and capabilities for the Sudanese dialect.
7. No code or data release plans are mentioned in the document. If possible, I suggest authors release their contribution.

---

### Meta-Review · Area_Chair_tcAa · 2023-04-06

**Recommendation:** Invite to revise
**Confidence:** 4

**Metareview:**

**Summary**

* The paper presents a trained XLM-RoBERTa model on a Sudanese-Arabic dialect. They used 6 million sentences to train the model.

**Strength**
* The work is done in a low-resourced dialect which is Sudanese.

**Weakness**
* Previous works on the Sudanese dialect exist (i.e. SudaBERT). And the comparison to that should be included in this paper instead of future work.
* The training and evaluation settings are weak. It would be good to evaluate the method on multiple tasks.
* Citations of other multi-lingual methods are missing.
* The codes / or data used are not released or no plans are mentioned in doing that.



**Summary:**

The paper presents a trained XLM-RoBERTa model on a Sudanese-Arabic dialect. They used 6 million sentences to train the model. Strength: the work is done in a low-resourced dialect which is Sudanese. Weakness: previous works on the Sudanese dialect exists and comparison to that would be useful to evaluate this models performance.

**Comments And Feedback To The Authors:**

The work is really interesting but data regarding the code/data and a valid comparison to existing methods would be great.

**Reason For Not Giving A Higher Recommendation:**

* The paper misses details. There needs to be compared with existing methods to evaluate the performance of the developed model. More, there's no link or plan discussed in the paper regarding the data/code. This would make the reproducibility of the work much more difficult.


**Reason For Not Giving A Lower Recommendation:**

N/A

---

### Decision · Program_Chairs · 2023-04-07

No revision received; not invited to archive